# Chronic Oleoylethanolamide Treatment Decreases Hepatic Triacylglycerol Level in Rat Liver by a PPARγ/SREBP-Mediated Suppression of Fatty Acid and Triacylglycerol Synthesis

**DOI:** 10.3390/nu13020394

**Published:** 2021-01-27

**Authors:** Adele Romano, Marzia Friuli, Laura Del Coco, Serena Longo, Daniele Vergara, Piero Del Boccio, Silvia Valentinuzzi, Ilaria Cicalini, Francesco P. Fanizzi, Silvana Gaetani, Anna M. Giudetti

**Affiliations:** 1Department of Physiology and Pharmacology “V. Erspamer”, Sapienza University of Rome, P.le Aldo Moro 5, 00185 Rome, Italy; adele.romano@uniroma1.it (A.R.); marzia.friuli@uniroma1.it (M.F.); silvana.gaetani@uniroma1.it (S.G.); 2Department of Biological and Environmental Sciences and Technologies, University of Salento, Via Prov.le Lecce-Monteroni, 73100 Lecce, Italy; laura.delcoco@unisalento.it (L.D.C.); serena.longo@unisalento.it (S.L.); daniele.vergara@unisalento.it (D.V.); 3Department of Pharmacy, University “G. d’Annunzio” of Chieti-Pescara, 66100 Chieti, Italy; piero.delboccio@unich.it (P.D.B.); silvia.valentinuzzi@unich.it (S.V.); 4Center for Advanced Studies and Technology (CAST), University “G. d’Annunzio” of Chieti-Pescara, 66100 Chieti, Italy; ilaria.cicalini@unich.it; 5Department of Medicine and Aging Science, University “G. d’Annunzio” of Chieti-Pescara, 66100 Chieti, Italy

**Keywords:** lipid metabolism, oleoylethanolamide, peroxisome proliferator-activated receptorγ, NMR spectroscopy, sphingolipids

## Abstract

Oleoylethanolamide (OEA) is a naturally occurring bioactive lipid belonging to the family of N-acylethanolamides. A variety of beneficial effects have been attributed to OEA, although the greater interest is due to its potential role in the treatment of obesity, fatty liver, and eating-related disorders. To better clarify the mechanism of the antiadipogenic effect of OEA in the liver, using a lipidomic study performed by ^1^H-NMR, LC-MS/MS and thin-layer chromatography analyses we evaluated the whole lipid composition of rat liver, following a two-week daily treatment of OEA (10 mg kg^−1^ i.p.). We found that OEA induced a significant reduction in hepatic triacylglycerol (TAG) content and significant changes in sphingolipid composition and ceramidase activity. We associated the antiadipogenic effect of OEA to decreased activity and expression of key enzymes involved in fatty acid and TAG syntheses, such as acetyl-CoA carboxylase, fatty acid synthase, diacylglycerol acyltransferase, and stearoyl-CoA desaturase 1. Moreover, we found that both SREBP-1 and PPARγ protein expression were significantly reduced in the liver of OEA-treated rats. Our findings add significant and important insights into the molecular mechanism of OEA on hepatic adipogenesis, and suggest a possible link between the OEA-induced changes in sphingolipid metabolism and suppression of hepatic TAG level.

## 1. Introduction

Oleoylethanolamide (OEA) is a naturally occurring bioactive lipid belonging to the family of N-acylethanolamides that has received great attention in the last two decades for its biological properties [1,2]. 

Diet-derived oleic acid promotes OEA formation in the small intestine of different species including rats and mice [3,4,5]; the membrane protein CD36, a multiligand class B scavenger receptor located on cell surface lipid rafts, plays a pivotal role in OEA biosynthesis by acting as a biosensor for food-derived oleic acid and facilitating OEA mobilization [6,7]. 

A variety of effects are attributed to exogenous administered OEA spanning in different domains, from neuroprotection [8,9,10] to memory [11], from inflammation to mood disorders [12,13,14], from the regulation of satiety to glucose homeostasis and lipid metabolism [8,9]. 

The majority of OEA’s biological functions explains its potential interest as a pharmacological target for the treatment of obesity and eating-related disorders [1,15,16]. Therefore, as a drug, OEA reduces food intake and body weight gain [3,17,18] in both lean and obese rodents. These effects are primarily related to the activation of the peroxisome proliferator-activated receptor-α (PPAR-α) [3,4,5,6,7], for which OEA shows high affinity. 

PPARα regulates several aspects of lipid metabolism [19], and in keeping with this, it was demonstrated that OEA, by recruiting PPARα, reduces serum cholesterol and triacylglycerol (TAG) levels and has beneficial effects on the high-fat diet-induced non-alcoholic fatty liver disease (NAFLD) in rats, by stimulating fatty acid β-oxidation, and by inhibiting lipogenesis [20]. 

OEA can influence sphingolipid metabolism in mice [21]. Indeed, OEA increases ceramide levels in cell cultures, via inhibition of ceramidase [22,23,24], an enzyme that catalyzes the degradation of ceramide to sphingosine and fatty acids. Ceramide and ceramide-derived sphingolipids are structural components of membranes and have been linked to insulin resistance, oxidative stress, inflammation [7,8,9] and then to hepatic steatosis. 

It was reported that ceramide can influence TAG homeostasis, and hepatic steatosis throughout PPARγ [19], a member of a nuclear hormone superfamily. 

Aberrant hepatic PPARγ expression can stimulate hepatic lipogenesis [25] and induce steatosis in mice hepatocytes [26,27], by up-regulating proteins involved in lipid uptake, and TAG storage such as CD36, monoacylglycerol O-acyltransferase 1, and stearoyl-CoA desaturase 1 (SCD1) [25,28].

PPARγ and CD36 mRNA expression are up-regulated in high-fat diet-induced liver steatosis in mice [29]. CD36 expression has been associated with insulin resistance in humans with type 2 diabetes [30,31] and increased hepatic *Cd36* gene expression was reported to increase fatty uptake, TAG accumulation [32,33] and fatty liver [32].

To clarify the mechanism of the antilipogenic effect of OEA in the liver, by using different approaches, such as ^1^H-NMR, LC-MS/MS and thin-layer chromatography (TLC), we performed a lipidomic analysis of the whole hepatic lipid composition. We found that OEA induced a significant decrease in ceramidase activity and profoundly impacted hepatic lipid composition by significantly increasing sphingomyelin, 24:0ceramide, dihydroceramide, and sphingosine with a concomitant significant decrease in glucosylceramide level. Moreover, a significant decrease in TAG level was also measured. We found that the antilipogenic effect of OEA was associated with a decreased activity and expression of the key enzymes involved in fatty acid and TAG syntheses, such as acetyl-CoA carboxylase (ACC), fatty acid synthase (FAS), diacylglycerol acyltransferase (DGAT), and SCD1. Moreover, we found that PPARγ and SREBP-1 protein expression was significantly reduced in the liver of OEA-treated rats. A possible link between OEA-induced alterations in sphingolipid metabolism and PPARγ-mediated suppression of enzymes involved in TAG synthesis was proposed.

## 2. Materials and Methods 

### 2.1. Animals, Diet and Chronic Treatments

Adult male Wistar-Han rats (250–300 g at the beginning of the study) were housed in single cages under controlled conditions of temperature and humidity (T = 22 ± 2 °C; 60% of relative humidity) and were kept on a 12 h light/dark cycle. Rats were fed for 11 weeks with a rodent diet (D12450B, Research Diets Inc., New Brunswick, NJ, USA) containing 3.82 kcal/g, which were distributed among carbohydrates, proteins, and fats according to the following percentages: 70%, 20%, and 10%. The rats were accustomed to handling and injections for 7 days before the beginning of the experiments. Rats, matched for body weight, were randomly divided into two different groups (12 rats per group) and housed individually in metal cages (30 × 30 × 30 cm). One group was treated with vehicle (VEH) and the other with OEA for 2 weeks. Both groups had free access to both food and water. 

Both OEA and VEH solutions were freshly prepared on each test day and administered about 10 min before dark onset by following our previous protocols [34,35,36]. OEA (Sigma-Aldrich, St. Louis, MI, USA) was prepared in the laboratory [37] and administered by intraperitoneal injection (i.p.) at the dosage of 10 mg kg^−1^ in a vehicle of saline/polyethylene glycol/Tween 80 (90/5/5, *v*/*v*). Theses dosage, vehicle and route of administration were chosen on the basis of the extensive scientific literature published by our and other research groups during the last 20 years. The i.p. route of administration was the most reliable to obtain a lower dosage variability, as compared to oral administration of OEA mixed in the rodent diet, and the highest bioavailability of OEA with the less stressful manipulation of the animals. Animal body weight was monitored on daily basis. As expected from previous observations [1], OEA-treated rats showed a significant decrease of body weight gain as compared to the VEH rats (Appendix A). 

At the end of the 2-week treatment period, animals were sacrificed, their livers immediately collected, washed in ice-cold phosphate-buffered saline, snap frozen in 2-metylbutane (−60 °C) and stored at −80 °C until analyzed. All experiments were carried out in accordance with the European directive 2010/63/UE governing animal welfare and with the Italian Ministry of Health guidelines for the care and use of laboratory animals. 

### 2.2. ^1^H NMR Spectroscopy

Hepatic total lipids were extracted using the Bligh and Dyer procedure. Lipid extracts (VEH and OEA samples) were analyzed by using 600 µL of deuterated chloroform (CDCl_3_) and transferred to a 5 mm NMR tube, using tetramethylsilane (TMS, δ = 0.00) as an internal standard. 1D 1H and 2D ^1^H J-resolved, ^1^H–^1^H COSY Correlation Spectroscopy, ^1^H–^13^C HSQC Heteronuclear, and ^1^H–^13^C HMBC, Multiple Bond Correlation, spectra were acquired at 300 K on a Bruker Avance III NMR spectrometer (Bruker, Biospin, Milan, Italy), operating at 600.13 MHz for ^1^H observation, equipped with a TCI cryoprobe incorporating a *z*-axis gradient coil and automatic tuning-matching (ATM). The following parameters were used for ^1^H NMR spectrum: 64 K data points, spectral width of 20.0276 Hz, 64 scans with a 2 s repetition delay, 90° power pulse (p1) 7.06 µsec and power level −8.05 dB. The acquisition and processing of spectra were performed using Topspin 3.5 software (Bruker Biospin, Milan, Italy). Resonances of fatty acids and metabolites were assigned according with data available in the literature [38,39,40].

### 2.3. Tissue Collection and Lipid Analysis 

For sphingolipid and neutral lipid analyses by TLC, 0.1 mg of tissue homogenate proteins was used. The developing system was composed of toluene:methanol (70:30, *v*:*v*) for sphingolipid analysis and hexane:ethyl ether:acetic acid (70:30:1, *v*:*v*) for neutral lipids. Ceramide and sphingomyelin were identified by comparison with specific standards (C18 ceramide and sphingomyelin (d18:1/12:0) from Avanti Polar). After separation, plates were sprayed uniformly with 8% cupric sulfate in 8% aqueous phosphoric acid, allowed to dry 10 min at room temperature, and then placed into a 145 °C oven for 10 min as reported in [41]. Band intensities were quantified using Image LabTM Version 6.0.1 2017 (Bio-Rad Laboratories, Inc., Segrate (MI)—Italy) software. 

Hepatic TAG level was also determined using an enzymatic colorimetric kit (RANDOX Laboratories), following manufactory instructions.

### 2.4. LC-MS/MS Analysis

Sphingolipid analysis was performed with LC-MS/MS following the previously described method [42]. Briefly, lipid extracts from hepatic homogenates were dried down and reconstituted in 1000 µL di CHCl_3_:CH_3_OH (2:1, *v*:*v*), vortexed and centrifuged for 15 min at +4 °C. The supernatant was subsequently diluted 1:10 with a solution formed by 50% solvent A (H_2_O with Formic acid 0.1%) and 50% solvent B (CH3OH:iPrOH:ACN (4:4:1, *v*:*v*) with Formic acid 0.1%. A volume of 10 µL of Internal Standard solution (Avanti Polar), containing d17So1P 0.1 µg/mL, C17Cer 0.01 µg/mL, C17GlcCer 1.0 µg/mL, was added to 90 µL of sample diluted as described above. After vortexing and centrifuging (5 min at +4 °C) 90 µL of the sample was transferred in vials for subsequent LC-MS/MS analysis.

The LC-MS/MS system consists of an HPLC Alliance HT 2795 Separations Module coupled to Quattro UltimaPt ESI tandem quadrupole mass spectrometer (Waters Corporation, Milford, MA, USA), operating in positive ion mode. The chromatographic separation of analytes was performed using Ascentis Express Fused-Core C18 2.7 µm, 7.5 cm × 2.1 mm column. In a total run time of 25 min, the elution of ceramides was achieved through a gradient of mobile phases, starting from 50% to 100% of methanol:isopropanol:acetonitrile (4:1:1, *v*:*v*) with Formic acid 0.1% (solvent B), water with Formic acid 0.1% was used as solvent A. The flow rate was 0.20 mL/min. The capillary voltage was 3.8 kV, source temperature was 120 °C, desolvation temperature was 400 °C, and the collision cell gas pressure was 3.62 × 10^−3^ mbar argon. Multiple reaction monitoring (MRM) functions for detection of sphingolipids are reported in Appendix A.

### 2.5. Liver Microsome Isolation and DGAT Activity Measurements

At the end of the experimental period, livers from VEH and OEA rats were removed and suspended in a medium containing 250 mM sucrose, 1.0 mM Tris-HCl (pH 7.4), 0.5 mM EGTA. In the same medium, the liver was gently homogenized with a Potter-Elvehjem homogenizer. Microsomes were isolated by differential centrifugation as in [43] with a final ultra-centrifugation step at 40,000× *g* for 1 h. The pellet of this centrifugation, corresponding to the microsomal fraction, was suspended in the same sucrose buffer. 

DGAT activity was measured on the microsomal fraction as described in [44] using endogenous diacylglycerols as substrates in the presence of [1-^14^C]palmitoyl-CoA (240 Bq/mol). The incubation was terminated after 1 min by the addition of 2 mL of methanol/chloroform (2/1, *v*/*v*). After lipid extraction, TAG were isolated by TLC on Silica G as reported in [44]. The silica, containing the TAG fraction, was scraped from the plate for radioactivity measurements.

### 2.6. Assay of Hepatic Enzymatic Activities 

For sphingomyelin synthase assay, the liver was homogenized in a buffer containing 50 mM Tris-HCl, 1 mM EDTA, 5% and sucrose and centrifuged at 2700× *g* for 10 min. The supernatant was then used for the activity assay. The mix reaction contained 50 mM Tris-HCl (pH 7.4), 25 mM KCl, C6-NBDceramide (0.1 mg/mL), and PC (0.01 mg/mL). The reaction was started with 100 µg proteins and the mixture was incubated at 37 °C for 2 h. Lipids were extracted in chloroform/methanol (2/1, *v*/*v*), dried under nitrogen, and separated by TLC. For the fluorescence measurement, plates were scanned with ChemiDoc™ MP System with Image Lab™ Software. 

For neutral and acid sphingomyelinase assay, the liver was homogenized in a buffer containing 50 mM Na acetate, 5 mM MgCl_2_, 1 mM ETA and 0.5% triton X-100, pH4.5. The enzymatic assay was conducted with a Colorimetric Sphingomyelinase Assay Kit furnished by MERK (Italy).

Ceramidase activity was measured as reported in [45], by incubating 25 µg of protein from liver homogenate with 100 µM N-lauroyl ceramide (Nu-Chek Prep) as substrate in assay buffer (125 mM sucrose, 0.01 mM EDTA, 125 mM Na acetate, and 3 mM DTT, pH 4.5) for 1h min at 37 °C. Reactions were stopped by the addition of a mixture of chloroform/methanol (2:1, *v*:*v*). The organic phases were collected, dried under N2, and the N-lauroyl amount was measured by LC-MS/MS.

ACC activity was determined as the incorporation of radiolabelled acetyl-CoA into fatty acids in a coupled assay with FAS reaction as described [46]. The reaction was carried out at 37 °C for 8 min. To determine FAS activity, malonyl-CoA was included in the ACC assay mixture, while adenosine triphosphate (ATP), butyryl-CoA, and FAS were omitted. The assay was allowed to proceed for 10 min. 

### 2.7. Western Blot Analyses

Proteins were extracted from the whole liver homogenate using RIPA lysis buffer (15 mM Tris-HCl, 165 mM-NaCl, 0.5% Na-deoxycholate, 1% Triton X-100 and 0.1% SDS), with a protease inhibitor cocktail (1:1000; Sigma-Aldrich, St. Louis, MI, USA) and 1 mM-PMSF (phenylmethanesuOEAnyl fluoride solution). Total protein levels of the lysate were determined using the Bradford method (Bio-Rad Laboratories). After boiling for 5 min, proteins were loaded and separated by SDS-polyacrylamide gel. The samples were then transferred to a nitrocellulose membrane (Bio-Rad Laboratories) and blocked at room temperature for 1 h using 5% (*w*/*v*) non-fat milk in TBS-Tris buffer (Tris-buffered saline (TBS) plus 0.5% (*v*/*v*) Tween-20, TTBS). The membranes were incubated overnight at 4 °C with primary antibodies against ACC (Cell Signaling #3676, Rabbit 1:1000), FAS (Cell Signaling #3180, Rabbit 1:1000), DGAT1 (Novus Biologicals #NB110-41487, Rabbit 1:1000), DGAT2 (Novus Biologicals #NBP1-71701, Mouse 1:1000), PPARγ (Santa Cruz #sc-7273, Mouse 1:500), sterol regulatory element-binding protein-1 (SREBP-1) (Santa Cruz #sc-365513, Mouse 1:500), CD36 (Santa Cruz #sc-7309, Mouse 1:1000), and SCD1 (Santa Cruz #sc-58420, Mouse 1:1000). β-actin (Cell Signaling #8457, Rabbit 1:1000), was used to determine loading fairness. Western blotting analyses were performed using Amersham ECL Advance Western Blotting Detection Kit (GE Healthcare, Little ChaOEAnt, UK) and detection was made using a VersaDoc Image System (Bio-Rad Laboratories, Hercules, CA, USA). β-actin was used to determine loading fairness. After washing with TTBS, the blots were incubated with peroxidase-conjugated monoclonal anti-rabbit secondary antibodies (Sigma-Aldrich, St. Louis, MI, USA) at 1:10.000 dilutions at room temperature for 1–2 h. The blots were then washed thoroughly in TTBS. Western blotting analyses were performed using Amersham ECL Advance Western Blotting Detection Kit (GE Healthcare, Little ChaOEAnt, UK). Densitometric analysis of the immunoblots was performed using Image Lab^TM^ Version 6.0.1 2017 (Bio-Rad Laboratories, Inc., Segrate (MI)—Italy) software.

### 2.8. Hepatic Cell Tretaments 

The human hepatocellular carcinoma cell line HLF was maintained in Dulbecco’s minimum essential medium eagle (DMEM) low glucose with 10% fetal bovine serum (FBS), 100 U/mL penicillin, 100 μg/mL streptomycin, and 2 mM glutamine. Cells were cultured at 37 °C with 5% partial pressure of CO_2_ in a humidified atmosphere. Cells were treated for 2 h with OEA or Carmofur (Cayman Chemical) at the concentration of 10 µM and 5 µM, respectively. OEA and Carmofur were dissolved in dimethyl sulfoxid (DMSO) at a 10 mM concentration. At the end of the incubation time total proteins were extracted using RIPA lysis buffer, with a protease inhibitor cocktail and 1 mM-PMSF. Total protein lysate levels were determined using the Bradford method. After boiling for 5 min, proteins were loaded and separated by SDS-polyacrylamide gel and probed with PPARγ antibody.

### 2.9. Multivariate Statistical Analyses 

The ^1^H-NMR spectra of lipid extracts (ZG Bruker pulse program experiments) were used for multivariate statistical analysis. Each spectrum was segmented in fixed rectangular buckets of 0.04 ppm width and successively integrated, by using Amix 3.9.13 (Bruker Biospin, Milano, Italy) software. The spectral regions between 7.45–7.00, 2.00–1.50 ppm, due to the residual peaks of solvents (chloroform and residual water), were discarded. The total sum normalization and the Pareto scaling procedure (performed by dividing the mean-centered data by the square root of the standard deviation) were then applied to the whole data to minimize small differences due to sample concentration and/or experimental conditions among samples [47]. Unsupervised Principal Component Analysis (PCA) and Orthogonal Partial Least Squares Discriminant Analysis (OPLS-DA) were applied to examine the intrinsic variation in the data using Metaboanalyst software [48]. The validation of statistical models was performed and further evaluated by using the internal 10-fold cross-validation and with the permutation test (100 permutations). Two parameters, R^2^ (the total variations in the data) and Q^2^ (the predictive ability of the models) were analyzed to describe the goodness of the statistical models [49]. The box and whisker plots, obtained for the discriminant metabolites found by multivariate analyses, summarize the normalized bucket values (box limits indicate the range of the central 50% of the data, with a central line marking the median value; the notch indicates the 95% confidence interval around the median of each group).

Metabolic results are reported as means ± standard error of the mean (SEM). The comparison between the two groups was made using Student’s t-test. Differences between groups were considered statistically significant when *p* < 0.05.

## 3. Results

### 3.1. ^1^NMR-Hepatic Lipidomic Analysis and Identification of Lipid Classes

An overall study of the lipid composition of hepatic homogenate from VEH and OEA rats was performed by ^1^NMR. 2D COSY, HSQC, HMBC and *J-resolved* spectra were randomly performed on samples and used to accurately assign the lipid classes present in samples (saturated, unsaturated fatty acids, phospholipids). The identified NMR signals of lipid extracts and related assignments are reported in Table 1 and Appendix A. 

The ^1^H NMR spectrum (Appendix A) can be divided into three broad regions: 3.0–0.65 ppm for fatty acids and sterol methyl and methylene resonances (A); 5.00–3.00 ppm for phospholipids head groups and glycerol backbone proton resonances (B); 6.00–5.00 ppm for vinyl protons resonances for fatty acids and sphingolipids (C) [38,50]. Two signals of cholesterol (CH_3_ groups) were identified in the spectral region between 1.05 and 0.5 ppm. In particular, distinctive singlets of the methyl groups of cholesterol are identified at 1.02 and 0.69 ppm. A characteristic contribution from the choline head groups was found in phosphatidylcholine (PC), phosphatidylethanolamine (PE) and sphingolipids (SL), with sharp singlet signals at approximately 3.35–3.15 ppm. In particular, the signals at 3.35–3.32 and 3.17 ppm of methyl groups (-N^+^(C*H*_3_)_3_) were ascribable to PC/PE and SM, respectively. Moreover, the presence of PC and PE were confirmed by resonances at 3.60 and 4.07 ppm, respectively, while the multiplets at 5.74 ppm were diagnostic of the characteristic sphingenine moiety vinyl protons (HO-CH-C**H**=C**H**-) and, therefore, suggested the presence of sphingolipids, including sphingomyelin, as also reported in the literature for specific systems [40,51,52]. All diacylglycerophospholipids (DAGP) are represented by the backbone glycerol sn-2 proton multiplet at ~5.21 ppm, while TAG show characteristic signals at 5.27, 4.29–4.27 and 4.15 ppm, partially overlapped with other moieties peaks. Among DAGP, phosphatidylglycerol (PG) signals appeared at 3.74 ppm [38,52]. Finally, the signal at 2.38 ppm appears for the presence of *n*-3 polyunsaturated fatty acids together with other NMR signals at 0.87, 1.30, 1.59 ppm [50].

### 3.2. Multivariate Analysis of VEH and OEA Liver Samples

A multivariate statistical approach was applied to the NMR data, without removing any of the observations as outliers, to reveal the general trend or data grouping of the samples. Unsupervised Principal Component Analysis (PCA) was performed to investigate the differences between samples, after the pre-processing treatment of the NMR spectra. In the PCA score plot of the corresponding model (Figure 1a), three principal components (t[1]/t[2]/t[3]) explained more than 80% of the total variance (R^2^X = 0.81, Q^2^ = 0.45).

OEA and VEH samples resulted well separated in the PCA score scatterplot, in particular when the t[2] and t[3] principal components were observed. The separation of OEA and VEH samples as two specific classes was further confirmed by the supervised PLS-DA analysis (Appendix A), which gave a satisfactory model (R^2^ = 0.95, Q^2^ = 0.81).

The OPLS-DA model (R^2^X = 0.79, R^2^Y = 0.98, Q^2^ = 0.90, Figure 1b), obtained from ^1^H NMR lipid extracts, showed a clear-cut separation of samples, which were clearly distinct along the predictive t[1] axis. The analysis of NMR signals responsible for the class separation was reported as box and whisker plots, which summarize the normalized values obtained for the buckets containing discriminant metabolite signals (Figure 1c). Relative increased values of phospholipids (such as PC/PE/SL), polyunsaturated fatty acids (PUFA), among which linoleic acid and cholesterol were found in OEA vs. VEH samples. Vice versa, a significantly decreased amount of TAG, and PG in OEA vs. VEH rats was observed.

### 3.3. Hepatic Lipid Analysis by TLC and LC-MS/MS

By NMR spectroscopy we identified specific characteristic sphingenine moieties, suggesting the possible presence of sphingomyelin [40,51,52]. On the basis of the NMR result, we investigated on the different sphingolipid species by TLC and LC-MS/MS analysis.

TLC analysis (Figure 2a) revealed a significant increase in the amount of both ceramide (Figure 2b) and sphingomyelin (Figure 2c) in OEA versus VEH rats.

To verify whether the changes in the hepatic level of ceramides and sphingomyelin was linked to changes in the activity of main enzymes involved in sphingolipid metabolism, we measured the activity of sphingomyelinase (neutral and acid forms), sphingomyelin synthase (Figure 2d) and ceramidase (Figure 2e). We found a significant decrease in the ceramidase activity in OEA vs. VHE animals. No significant changes were instead measured in either neutral and acid sphingomyelinase or sphingomyelin synthase activities.

Moreover, by TLC analysis we also evaluated the level of neutral lipids (Figure 2f) in the liver of OEA vs. VHE rats. According to the NMR analysis, a reduced level of TAG and increased level of cholesterol was found in the liver of OEA-treated rats with respect to VHE. No significant changes were instead measured in the level of cholesterol esters, free fatty acids, and diacylglycerols (Figure 2g). By using an enzymatic colorimetric assay, we also measured the amount of total hepatic TAG in VEH and OEA groups of rats. We found a reduced (about 40%) hepatic amount of TAG in OEA versus VEH animals (0.074 ± 0.01 mg/mg protein in VHE versus 0.044 ± 0.01 mg/mg protein in OEA; *p* < 0.05) (Figure 2h). These results confirmed data obtained by NMR analysis.

To gain insight into the molecular mechanism of the OEA effect on sphingolipid pathway (Figure 3a), using LC-MS/MS we measured the hepatic concentrations of long (C16-C20) and very long (≥C24) chain ceramides as well as of their precursors, dihydroceramides. We observed a significant accumulation of both dihydroceramide (d18:0/16:0) (C16dHCer) and C24:0ceramide in OEA as compared to VHE (Figure 3b). Sphingosine, the bioactive amino-alcohol backbone of sphingolipids, was up-regulated in the liver of OEA rats as well. Moreover, the amount of C16glucosilceramide showed a decrease in OEA with respect to VHE.

### 3.4. Activity and Expression of Triacylglycerol and Fatty Acid Synthesis Enzymes

In the liver, TAG synthesis is catalyzed by two main DGAT isoforms, DGAT1 and DGAT2. Although both DGAT enzymes synthetize TAG from fatty-acyl-CoA and diacylglycerols, they have different roles and regulatory mechanisms [43].

We investigated whether OEA chronically administered to rats might impact the activity and the expression of hepatic DGAT enzymes.

Western blot analyses (Figure 4a) revealed a significant decrease of both hepatic DGAT1 and DGAT2 expressions in OEA- with respect to VEH-treated rats (Figure 4b). Moreover, DGAT activity, measured in the liver microsomal fraction, was significantly decreased in OEA with respect to VEH rats (Figure 4f).

SCD1 synthesizes oleic acid from stearate, a conversion that facilitates the biosynthesis of TAG and other neutral lipids. SCD1 expression is highly correlated with liver steatosis [53]. Western blot analysis (Figure 4a) revealed that the hepatic protein level of SCD1 was significantly lower in OEA versus VEH rats (Figure 4c).

DGAT can catalyze TAG synthesis by using de novo synthetized fatty acids or fatty acids taken-up from the bloodstream. The de novo synthesis of fatty acids is catalyzed by ACC, which uses the ATP-dependent carboxylation of acetyl-CoA to produce malonyl-CoA and FAS, which catalyzes the synthesis of palmitoyl-CoA, using malonyl-CoA as substrate.

ACC and FAS protein expressions and activities were analyzed in the liver of VEH and OEA rats. The result obtained by western blotting (Figure 4a) demonstrated that both ACC and FAS protein levels were statistically reduced in OEA versus VEH (Figure 4d). This data was consistent with a significantly reduced activity of both ACC and FAS measured in OEA versus VEH rats (Figure 4f).

Moreover, we also measured, by western blot analysis (Figure 4a) the hepatic level of CD36, a protein that facilitates the transport of long-chain fatty acids in several cell types [54]. We found that the hepatic protein level of CD36 decreased about 2-fold in OEA versus VEH rats (Figure 4e).

### 3.5. Hepatic PPARγ and SREBP-1 Protein Expression

PPARγ regulates several proteins associated with fatty acid synthesis and TAG storage [19]. Moreover, SREBP-1 represents the master transcription factor regulating lipogenic enzyme expression in the liver [55]. We assessed whether the down-regulation of the lipogenic proteins upon OEA treatment could depend on PPARγ and/or SREBP-1c.

By western blot analysis (Figure 5a) we measured the PPARγ and SREBP-1 hepatic protein content in both VEH and OEA groups of rats. We found a significantly decreased protein level of both PPARγ (Figure 5b) and SREBP-1 (Figure 5c) in the liver of OEA rats with respect to VEH rats. The suppressive effect of OEA on PPAγ expression was also demonstrated in vitro by incubating HLF hepatic cells for 2 h with 10 µM OEA (Figure 5d,e). Moreover, the ceramidase inhibitor Carmofur, which increases OEA cell level by inhibiting fatty acid hydrolases [56], induced both alone and in association with OEA, a strong decrease in PPARγ expression (Figure 5d,e).

## 4. Discussion

In the present study, we demonstrated that chronic OEA treatment was able to affect hepatic sphingolipid metabolism, and to significantly decrease TAG level with concurrent reduction of PPARγ and SREBP-1 protein expression.

Lipidomic analysis of the whole hepatic lipid component showed that OEA increased both ceramide and sphingomyelin content. It is particularly noteworthy the increased amount of C24:0ceramide induced by OEA, considering the positive correlation recently reported between the increased plasmatic level of C24:0ceramide and decreased cardiovascular events [57], and the link between C24-ceramide, exogenously administered and the reduction in PPAR-γ expression [58,59]. It is also worth considering the decreased glucosylceramide amount measured in the liver after OEA administration, considering that decreased glucosylceramide synthesis was reported to ameliorate insulin sensitivity and decrease TAG accumulation [60].

It is well known that sphingolipid metabolism is subject to very complex regulatory mechanisms and in this context, sphingosine plays a pivotal role. An increase of sphingosine after OEA treatment is apparently in contrast with inhibitory role of OEA toward ceramidase. However, it must be considered that sphingosine can derive also from sphingosine-1P throughout a phosphatase activity [61], or by a salvage pathway by the breakdown of complex lipids [62]. Overall, our preliminary data open to future lipidomic studies aimed at analyzing quantitatively and comprehensively all these lipid species.

In our study, we observed that OEA-induced a significant decrease of PPARγ expression both in vivo and in vitro. In this latter system the link between OEA and PPARγ was reinforced by using Carmofur, a fatty acid hydrolase inhibitor that can block OEA degradation and, thus, increase OEA level in the cell [56].

A large number of recent studies suggest that PPARγ might represent a relevant clinical target for the treatment of NAFLD [63] and another set of data propose a relationship between sphingolipid biosynthesis and NAFLD [62]. For the first time, we found that OEA decreased the hepatic expression of PPARγ and of its downstream target protein CD36, an effect that in turn, inhibits TAG accumulation.

We overall consider that the hepatic reduction of PPARγ expression was linked to a reduction in TAG content and this observation is in keeping with previous studies demonstrating that a reduced PPARγ expression in the liver is associated with hepatic lipogenesis and TAG content [64].

We confirm our hypothesis by investigating the impact of chronic administration of OEA on the activity and the expression of TAG and fatty acid synthesis enzymes.

The final step in TAG synthesis is catalyzed by DGAT isoforms, DGAT1 and DGAT2 [65] with distinct protein sequences and different biochemical, cellular, and physiological functions [52]. PPARγ is involved in the regulation of both DGAT1 and DGAT2 enzyme expression, and interestingly knockdown of hepatic PPARγ was reported to reduce hepatic lipid accumulation and both DGAT1 and DGAT2 expressions [64,65,66,67].

We found that DGAT1 and DGAT2 expression and total DGAT activity decreased in the livers of OEA-treated rats. OEA treatment significantly decreased also the hepatic level of SCD1, a member of the fatty acid desaturase family that catalyses the conversion of stearoyl-CoA to oleoyl-CoA, a major substrate for TAG synthesis [53]. All these data are in line with the lower level of TAG we measured in the liver of OEA-treated rats (Figure 2).

SREBP-1c is a transcription factor that activates the expression of most genes required for hepatic lipogenesis, such as ACC, FAS, and SCD1 [68,69]. Increased levels of SREBP-1c mRNA were demonstrated in livers of several mouse models characterized by insulin resistance and increased rates of hepatic lipogenesis [69].

Li et al. [20] reported that OEA inhibits hepatic de novo lipogenesis throughout a SREBP-1-mediated inhibition of SCD1, and ACC mRNA expression [20]. According to that study, we found that OEA decreases the expression of SREBP-1. Moreover, we observed that OEA decreases ACC, FAS, and SCD1 protein expression, and ACC and FAS activities, thus suggesting a possible link between the decreased level of ACC, FAS, and SCD1 and the down-regulation of SREBP-1.

It is well known that polyunsaturated fatty acids (PUFA) can suppress hepatic SREBP-1 nuclear abundance and the expression of its hepatic target genes [70,71,72]. Considering that NMR analysis of the hepatic lipid composition highlighted an increased level of PUFA in the liver of OEA-treated rats, we can speculate that the down-regulation of SREBP-1 could be associated with the altered lipidomic profile. About the mechanism responsible for the OEA-induced increase in hepatic PUFA, we hypothesize to be a mere consequence of the suppressed de novo lipogenesis that enriches the cell in saturated fatty acids.

Moreover, it should be also considered that NMR analysis showed a significantly increased level of PC in OEA-treated rats (Figure 1c): interestingly, a study reported that a chronically low PC level can drive TAG production regulating SREBP-1 processing [73]. Thus, according to our results, we cannot exclude a similar mechanism occurring also in the liver of OEA-treated animals.

Taken together, our results propose a dual role played by OEA on reducing hepatic TAG level: by recruiting the SREBP-1 system OEA might regulate key proteins involved in fatty acid synthesis, whereas by affecting PPARγ, it might influence fatty acid uptake, binding, and transport.

While the OEA-induced SREBP-1 down-regulation may be almost expected and essentially linked to changes in fatty acid profile, or probable modification in the OEA-induced insulin signaling [74], the PPARγ effect is rather new and deserves deeper investigation.

## 5. Conclusions

Taken together, our results describe the antiadipogenic functioning of OEA in the liver, which involves not only the *de novo* synthesis of fatty acids and the sphingolipid metabolism but also the transport of fatty acids into the liver and the synthesis of triacylglycerols. These findings add significant and important insights into the molecular mechanism of OEA on hepatic adipogenesis, suggesting an important role of PPARγ in regulating such process and further support the protective role of OEA in fatty-associated liver diseases.

## Figures and Tables

**Figure 1 nutrients-13-00394-f001:**
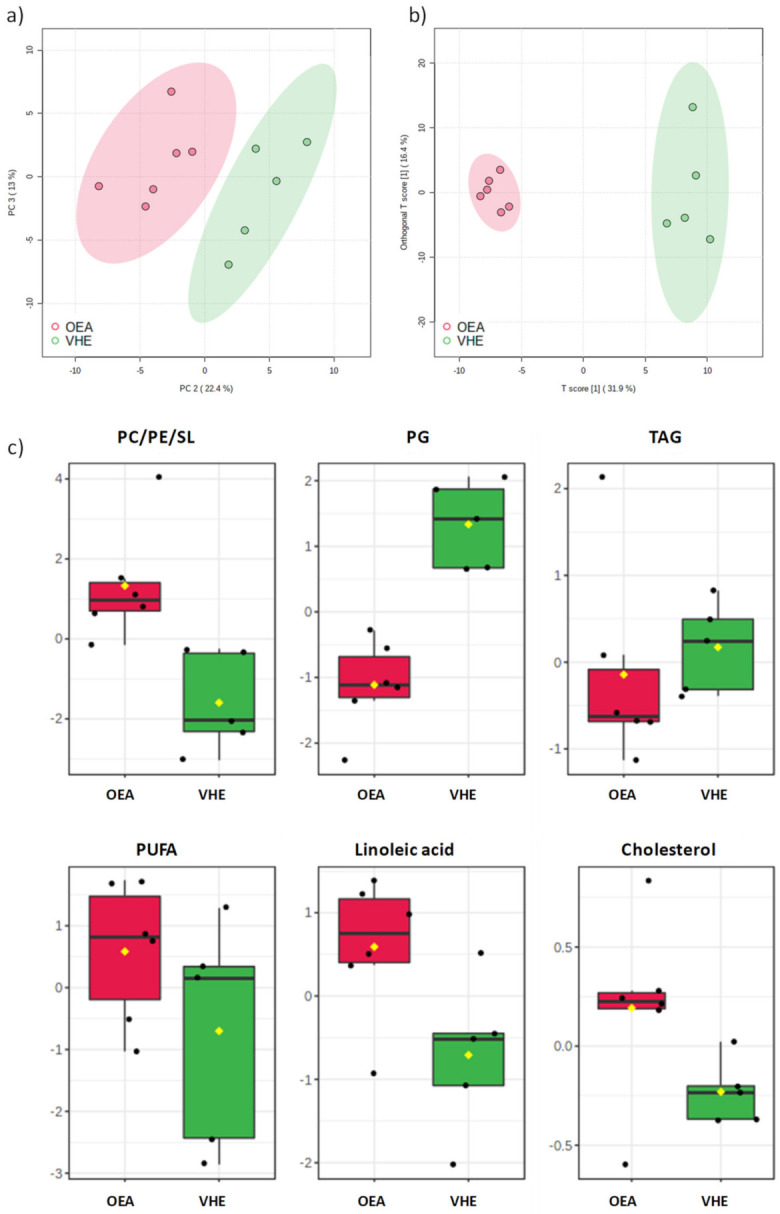
(**a**) Principal component analysis (PCA) score plot for OEA and VEH samples. (**b**) t[1]/to[1] orthogonal partial least squares discriminant analysis (OPLS-DA) score scatter plot for OEA and VEH samples, and (**c**) discriminant metabolites between the two groups are reported in the corresponding box and whisker plots, which summarize the normalized bucket values. The mean value for each group is indicated with a yellow diamond, while the notch indicates the 95% confidence interval around the median of each group, with dots placed past the line edges to indicate outliers. PC = phosphatidylcholine; PE = phosphatidylethanolamine; SL = sphingolipids; PG = phosphatidylglycerol; TAG = triacilglycerols; PUFA = polyunsaturated fatty acids.

**Figure 2 nutrients-13-00394-f002:**
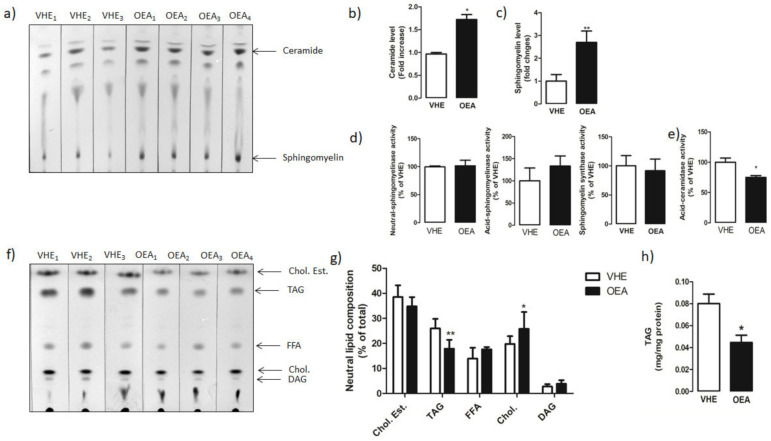
Hepatic sphingolipid and neutral lipid analysis. Total lipids from VEH and OEA groups were extracted and separated by TLC. (**a**) Representative TLC separation of sphingolipids. (**b**,**c**) Quantification of ceramide and sphingomyelin by densitometry analysis. Ceramide and sphingomyelin level is reported as fold change of OEA versus VHE. (**d**) Neutral and acid sphingomyelinase and sphingomyelin synthase activities reported as percentages of values measured in VEH. (**e**) Ceramidase activity assayed in liver homogenates. (**f**) Representative TLC separation of neutral lipids. (**g**) Neutral lipids are represented as percentage of the total. (**h**) Triacylglycerol (TAG) amount was spectrophotometrically quantified with a specific enzymatic assay. In the figure, the mean ± SEM of values obtained from three different analyses is reported. ** *p* < 0.005; * *p* < 0.05. Abbreviations: Chol. Est. = cholesterol ester; TAG = triacylglycerols: FFA = free fatty acids; DAG = diacylglycerols.

**Figure 3 nutrients-13-00394-f003:**
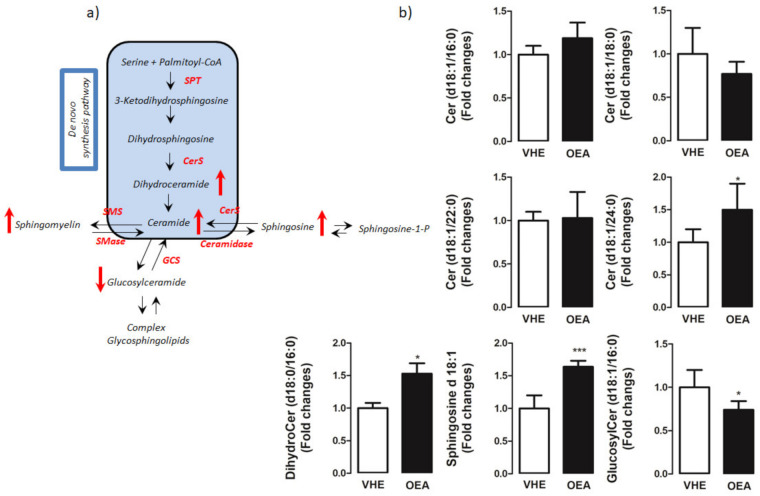
Hepatic sphingolipid analysis by LC-MS/MS. (**a**) Schematic representation of the sphingolipid pathway. De novo sphingolipid synthesis starts with the condensation of serine and palmitoyl-CoA by the rate-limiting enzyme, serine palmitoyltransferase (SPT) to 3-keto-dihydrosphingosine. Following further modification to dihydrosphingosine, ceramide synthases (CerS) convert dihydrosphingosine to dihydroceramide which is then desaturated to generate ceramides. Ceramides can be glycosylated by glucosylceramide synthase (GCS) to glucosylceramides which can serve as an intermediary for other glycosphingolipids, used as a substrate for sphingomyelin synthesis by sphingomyelin synthase (SMS) or deacylated by ceramidases to form sphingosine and subsequently, through the action of sphingosine kinases generate sphingosine-1- phosphate. (**b**) Distribution of sphingolipid species in the liver. In the figure, the mean ± SEM of values obtained from five different samples is reported; *** *p* < 0.001; * *p* < 0.05. Red arrows indicated metabolites that are up and down-regulated in OEA vs. VHE rats.

**Figure 4 nutrients-13-00394-f004:**
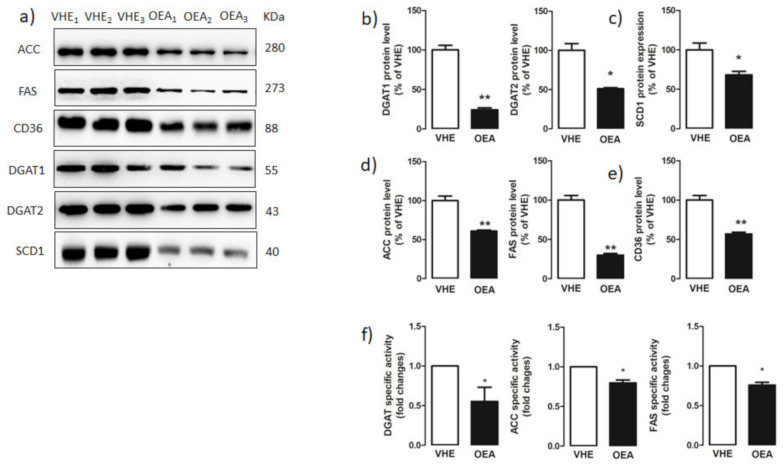
Hepatic fatty acid and triacylglycerol synthesis enzymes. (**a**) Proteins extracted from hepatic VEH and OEA samples were subjected to SDS-page electrophoresis. Membranes were analyzed by immunoblotting using specific antibodies for acetyl-CoA carboxylase (ACC), and fatty acid synthase (FAS), CD36, diacylglycerol acyltransferase 1 (DGAT1), DGAT2, and stearoyl-CoA desaturase 1 (SCD1). Each blot was normalized to the proper specific β-actin. (**b**–**e**) Blot signals were quantified by densitometric analysis and reported as % of the vehicle (VHE). (**f**) DGAT, ACC and FAS specific activities were assayed as reported in the Material and Methods section and reported as fold changes with respect to values measured in VEH rats. Values are the mean ± SEM of five different experiments. * *p* < 0.05; ** *p* < 0.005.

**Figure 5 nutrients-13-00394-f005:**
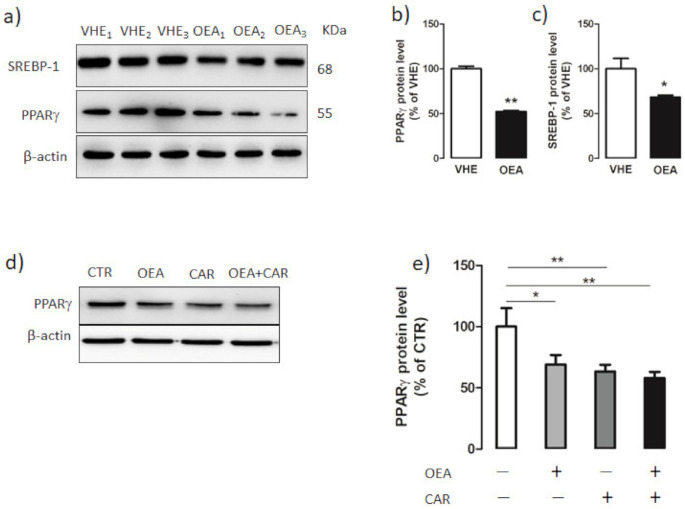
Hepatic PPARγ and SREBP-1 protein expression. (**a**) Hepatic proteins from VEH- and OEA-treated rats were subjected to SDS-page electrophoresis. Membranes were then subjected to immunoblotting using specific antibodies for sterol regulatory binding protein-1 (SREBP-1) and peroxisome proliferator-activated receptor γ (PPARγ). (**b**,**c**) Signals were quantified by densitometric analysis and expressed as percent of the vehicle (VHE). (**d**,**e**) PPARγ expression in HLF cells treated for 2 h with 10 µM OEA, 5 µM Carmofur (CAR) and OEA + Carmofur. β-actin was used as the loading control. Blot signals were quantified by densitometric analysis and reported as % of control (CTR) which was represented by cells without any treatment. Values are the mean ± SEM of five different experiments. * *p* < 0.05; ** *p* < 0.005.

**Table 1 nutrients-13-00394-t001:** Different lipid classes, with the corresponding resonance assignments, were identified by the NMR analysis. DAGP = diacylglycerophospholipids; FA = fatty acid chain; MUFA = monounsaturated fatty acids; SL = sphingolipids; PC = phosphatidylcholine; PG = phosphatidylglycerol; PUFA = polyunsaturated fatty acids; TAG = triacylglycerol; UFA = unsaturated fatty acids.

Resonance	^1^H NMR Signal	Chemical Shift (ppm)	Lipid Class
1	–C18*H*_3_	0.69	Cholesterol
2	–C*H*_3_	0.89	Total FA
3	–C19*H*_3_	1.02	Cholesterol
4	–(C*H*_2_)_n_–	1.25	Total FA
5	=CHC*H*_2_CH_2_(CH_2_)–	1.30	Total FA
6	–CO-CH_2_C*H*_2_–	1.62	Total FA
7	–C*H*_2_HC=	2.02	UFA
8	–C*H*_2_HC=	2.07	UFA
9	–CO-C*H*_2_–	2.37–2.23	Total FA
10	=CHC*H*_2_CH=	2.74	*n*-6 PUFA
11	=CHC*H*_2_CH=	2.88	*n*-3 PUFA
12	–N^+^(C*H*_3_)_3_	3.35–3.15	PC/PE/SL
13	CH_2_CHCH_2_	3.74	PG
14	PC and DAGP signals	4.5–3.5	PC
15	>C_2_*H* in glycerol backbone	5.21	DAGP
16	>C_2_*H* in glycerol backbone	5.26	TAG
17	–*H*C=C*H*– in fatty acyl chain PUFA and MUFA	5.42–5.29	UFA
18	OH-CH-C*H*=C*H*-	5.74	SL

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
