# Peer review of "Chronic Oleoylethanolamide Treatment Decreases Hepatic Triacylglycerol Level in Rat Liver by a PPARγ/SREBP-Mediated Suppression of Fatty Acid and Triacylglycerol Synthesis"

_nutrients, 2021, doi:10.3390/nu13020394_

Round 1

Reviewer 1 Report

The authors Romano et al. have done substantial experimental work to improve their manuscript. I am satisfied with it for the most part. However, this is only partially true for the newly added LC-MS/MS data. These raise more questions for me rather than providing answers or supporting the TLC data. The following points are inconclusive:

  1. the authors claim OEA treatment increases ceramides in rat liver, relying on the TLC data. The much more specific LC-MS/MS data show only a slight, probably borderline significant increase for C24 Cer. The summed total Cer contents would probably not be significantly different. What about C24:1 Cer, the most abundant species in the liver along with C16 and C24 Cer?
  2. By TLC, significant effects were also detected for sphingomyelin. Why were individual species not determined by LC-MS/MS?
  3. Furthermore, I have serious doubts about the correctness of the sphingosine data. The levels in the VHE control are about 10-fold higher than the levels of C16 Cer. Sphingosine should be present at much lower concentrations than ceramides. There is a lot of literature evidence on this.
  4. Moreover, at first sight, an increase of sphingosine after OEA treatment is contra-intuitive since inhibition of ceramidase activity by OEA on the one hand should increase ceramides and decrease sphingosine on the other. However, sphingolipid metabolism is subject to very complex regulatory mechanisms, thus a prediction is not always trivial. However, the authors should comment on this aspect.
  5. The decision to present C16:1 dihydroCer seems arbitrary, probably because it was the only species with a significant effect. However, the meaning of these data is close to zero. First, the nomenclature is incorrect. Dihydroceramides have by definition a d18:0 backbone. Then, C16:1 Cer and thus its precursor C16:1 dihydroCer are minor species. Therefore, I also doubt that in the VHE control C16:1 dihydroCer is as abundant as C24 Cer (both around 0.5 ng/ml), one of the most abundant ceramides in liver tissue.

Although I appreciate that the authors put a lot of effort and work into including LC-MS/MS sphingolipidomics data in their manuscript, I hope that my criticisms are met with understanding. The data MUST be thoroughly reviewed again before publication. Furthermore, I recommend listing the MRM transitions to measure sphingolipid species in the supplements.

Author Response

Reviewer 1.

The authors Romano et al. have done substantial experimental work to improve their manuscript. I am satisfied with it for the most part. However, this is only partially true for the newly added LC-MS/MS data. These raise more questions for me rather than providing answers or supporting the TLC data. The following points are inconclusive:

  1. the authors claim OEA treatment increases ceramides in rat liver, relying on the TLC data. The much more specific LC-MS/MS data show only a slight, probably borderline significant increase for C24 Cer. The summed total Cer contents would probably not be significantly different. What about C24:1 Cer, the most abundant species in the liver along with C16 and C24 Cer?
  • We thank the reviewer for his timely review, which certainly aims at improving our work. It should be considered that TLC provides a sum of ceramide species present in the sample. Unfortunately, the method established by Dr. del Boccio, and validated in a recent publication (Granzotto et al. Aging 2019, 11: 6336-6357), has been developed for the analysis of a small set of ceramide isoforms. Setting up a protocol for the analysis of other ceramide species requires additional time and the purchase of other standards which we are currently unable to do.

  1. By TLC, significant effects were also detected for sphingomyelin. Why were individual species not determined by LC-MS/MS?
  • We thank the reviewer for the note. However, in the present work, we focused our attention primarily on ceramides due to the reported effect of these molecules on PPARg Furthermore, following the previous suggestion, which we greatly appreciated, we also analyzed, by LC-MS/MS, the different species of ceramide. The analysis of sphingomyelin will require further optimization of the protocol including additional costs. This is a valid observation that we share and which requires further work in the future.

  1. Furthermore, I have serious doubts about the correctness of the sphingosine data. The levels in the VHE control are about 10-fold higher than the levels of C16 Cer. Sphingosine should be present at much lower concentrations than ceramides. There is a lot of literature evidence on this.
  • We agree with the reviewer's note, and according with this, also considering that the fundamental data is the different quantity found between VHE and OEA rather than absolute values, we decided to report the data in the form of fold changes. Previous published papers in the field of LC-MS and lipidomics support the validity of our analytical method and we are absolutely confident of these results. It is true that sphingosine should be present at much lower concentrations than ceramides but it is also true that only a complete analysis of all lipid species can really provide a clear indication of this difference. This can only be done accurately by a shot-gun approach. We therefore believe that a fair compromise to this problem is solved by expressing the data as relative, rather than absolute.

The limitation of our LC-MS/MS analysis has been now highlighted in the discussion section.

  1. Moreover, at first sight, an increase of sphingosine after OEA treatment is contra-intuitive since inhibition of ceramidase activity by OEA on the one hand should increase ceramides and decrease sphingosine on the other. However, sphingolipid metabolism is subject to very complex regulatory mechanisms, thus a prediction is not always trivial. However, the authors should comment on this aspect.
  • We agree with the reviewer that the metabolism of sphingolipid is a challenge. About the sphingosine increase after OEA treatment, despite ceramidase inhibition, it should be considered that sphingosine can also arise from sphingosine-1P throughout S1P-phosphatase (Mao C. et al. Biochim Biophys Acta. 2009). This explanation agrees with reports indicating that sphingosine-1P induces TAG accumulation in the liver through a PPARg-mediated mechanism (Jinbiao C. et al. Biocim. Biophys Acta 2016). Moreover, sphingosine can also derive from the breakdown of complex sphingolipids in the salvage pathway (Mangesh Pagadala et al. Trends Endocrinol Metab. 2012). Importantly, it has been reported that the detection of an increase in free sphingosine not only indicates enhancement of sphingolipid catabolism, but also raises the possibility of facilitated sphingosine salvage for ceramide synthesis (Kitatani et al. Cell Signal. 2008). This aspect has been now discussed in the revised version of the manuscript, thanks to the reviewer’s suggestion.

  1. The decision to present C16:1 dihydroCer seems arbitrary, probably because it was the only species with a significant effect. However, the meaning of these data is close to zero. First, the nomenclature is incorrect. Dihydroceramides have by definition a d18:0 backbone. Then, C16:1 Cer and thus its precursor C16:1 dihydroCer are minor species. Therefore, I also doubt that in the VHE control C16:1 dihydroCer is as abundant as C24 Cer (both around 0.5 ng/ml), one of the most abundant ceramides in liver tissue.
  • We apologize for the mistake. Now we have corrected the nomenclature of the dihydroceramide species.

Although I appreciate that the authors put a lot of effort and work into including LC-MS/MS sphingolipidomics data in their manuscript, I hope that my criticisms are met with understanding. The data MUST be thoroughly reviewed again before publication. Furthermore, I recommend listing the MRM transitions to measure sphingolipid species in the supplements.

  • We thank the reviewer for his constructive criticisms, which prompted us to improve our work. Overall, we believe that this LC analysis presents a carefully generated small data set, which would be a valuable resource for future analysis investigating, by a shot-gun approach, the effect of OEA on lipid metabolism.
  • As suggested, MRM transitions data have been included in the table S1.

Reviewer 2 Report

The response to my comments is not satisfactory. Specifically, in Fig. 3, actin cannot be used as loading control for all the proteins shown because the bands of the protein analyzed and those of actin are not coming from the same gel. Showing the protein bands derived from one gel and the actin bands derived from another gel to prove that the same amount of protein was loaded constitutes data falsification and is grounds for rejection. You can only make that statement if the protein and actin bands are derived from the same gel. Therefore, actin bands must be removed from Fig. 3 or the membranes shown should be paired with the actin bands derived from the same gel.

Author Response

Reviewer 2

The response to my comments is not satisfactory. Specifically, in Fig. 3, actin cannot be used as loading control for all the proteins shown because the bands of the protein analyzed and those of actin are not coming from the same gel. Showing the protein bands derived from one gel and the actin bands derived from another gel to prove that the same amount of protein was loaded constitutes data falsification and is grounds for rejection. You can only make that statement if the protein and actin bands are derived from the same gel. Therefore, actin bands must be removed from Fig. 3 or the membranes shown should be paired with the actin bands derived from the same gel.

  • We thank the reviewer for the note. Accordingly, we deleted the line relative to the β-actin from Figure 4 and we specified in the legend “Each blot was normalized to the proper specific β-actin”.

Round 2

Reviewer 1 Report

I accept the changes and the answers.

Only the phrase "sphingolipids and ceramides", which appears several times, should be corrected, since ceramides of course belong to the sphingolipids.

Reviewer 2 Report

The response to my comments was satisfactory. Recommend acceptance.

This manuscript is a resubmission of an earlier submission. The following is a list of the peer review reports and author responses from that submission.

Round 1

Reviewer 1 Report

Romano et al. investigated the hepatic lipidome by NMR and TLC analysis in a study in rats chronically administered the antiadipogenic bioactive lipid OEA. They found decreased liver TAG levels in OEA-treated rats, while Cer and SM were elevated. Moreover, they could show that levels and activities of enzymes central to TAG biosynthesis (e.g. DGAT1/2, ACC, FAS) were decreased in response to OEA treatment. Lastly, they observed reduced hepatic expression of PPARγ and SREBP-1 protein in rats administered OEA. Although the study contains interesting findings that are of potential interest to the community, the authors were not able to present direct causalities. Rather, the authors are tempted to speculate. The novelty and significance of the presented data is therefore limited. Furthermore, the methodology of lipidomics analyses is not state-of-the-art and should be improved.

Major points:

Line 99: Given that OEA is already used as an oral supplement, the authors should comment on why they used i.p. application in their rat study.

Line 107f: Was the efficiency of lipid extraction checked prior to NMR analysis? In LC-MS-based lipidomics approaches for instance (more frequently used than NMR-based), internal standards are usually added to the extraction and measured simultaneously with the analytes in the final sample. How was recovery determined in the present study? This may be decisive in a comparative lipidome study.

Line 119f: The description of the sphingolipid analysis should be made more precise. Which equal amounts of protein? Which different sphingolipid classes were separated? Which specific standards were used?

Line 148f: Why did the authors only determine the activity of neutral SMase, but not of acid SMase?

Line 198f: In my opinion, the major limitation of this study is the technique used to determine the hepatic lipidome. Although the NMR analyses performed appear to be accurate, identification of individual lipid classes (that may comprise several hundreds of subspecies) based on chemical shifts of individual structural elements is outdated and not very convincing. Especially because no subspecies could be determined, which e.g. in the case of sphingolipids (fatty acid side chains of different lengths) may exert different biological effects. In general, section 3.1 does not add any novel findings to the field of hepatic lipids.

Table 1, line 18: The assigned structural element is also present in e.g. ceramides and sphingosine. How was SM assigned to this signal?

Line 257: It was not clear to me how the NMR-based semi-quantification was performed. What was the reference (mg tissue, mg protein, another non-lipid signal,..)? Please clarify.

Figure 2: The authors should also determine the ceramidase activity to verify that the observed ceramide increase is due to the described OEA-mediated ceramidase inhibition. Without these data, also an OEA-related stimulation of the de novo synthesis could have taken place (which could also have been investigated in the microsomal fractions). What about DAGs? As they are by-products of SM synthesis, their contents would be of interest as well.

Figure 3: An interesting finding in my eyes is the significant reduction of FAS level and activity after OEA treatment. As described by the authors, FAS catalyzes the formation of palmitoyl-CoA. Palmitoyl-CoA is needed to build the sphingoid backbone of sphingolipids (de novo). Furthermore, it is incorporated as a fatty acid side chain in C16:0 Cer and C16:0 SM, abundant subspecies in the liver. The reduction of FAS could therefore rather indicate a reduction of ceramides and subsequent metabolites (such as SM, especially C16:0 subtypes). The opposite appears to be the case. This makes subspecies-resolved lipid analysis even more important, which the authors should aim for in order to improve the manuscript.

Lines 334-335: As mentioned above, this should be empirically confirmed.

Lines 338-350: No experimental data are presented that show direct interconnection between OEA treatment, elevated Cer levels and reduced PPARy expression. This paragraph thus is pure speculation and must be re-written. Preferably, the authors clarify the connection by additional experiments. 

Minor points:

Include the specie studied in abstract and title.

Figure 1: Quality/visibility should be improved.

Figure 2: Please explain LFV and LFO.

Figure 4: CD36 is not shown, but stated in the caption.

Line 321: “we found”

Author Response

Reviewer 1

Romano et al. investigated the hepatic lipidome by NMR and TLC analysis in a study in rats chronically administered with the antiadipogenic bioactive lipid OEA. They found decreased liver TAG levels in OEA-treated rats, while Cer and SM were elevated. Moreover, they could show that levels and activities of enzymes central to TAG biosynthesis (e.g. DGAT1/2, ACC, FAS) were decreased in response to OEA treatment. Lastly, they observed reduced hepatic expression of PPARγ and SREBP-1 protein in rats administered OEA. Although the study contains interesting findings that are of potential interest to the community, the authors were not able to present direct causalities. Rather, the authors are tempted to speculate. The novelty and significance of the presented data is therefore limited. Furthermore, the methodology of lipidomics analyses is not state-of-the-art and should be improved.

Major points:

Reviewer-Line 99: Given that OEA is already used as an oral supplement, the authors should comment on why they used i.p. application in their rat study.

Authors: We thank the reviewer for his/her comment. We know that OEA is on the market in USA as a supplement for the control of body-weight and that it is an oral formulation (capsules). However, by following our previous studies (for example see Romano A, Micioni Di Bonaventura MV, Gallelli CA, Koczwara JB, Smeets D, Giusepponi ME, De Ceglia M, Friuli M, Micioni Di Bonaventura E, Scuderi C, Vitalone A, Tramutola A, Altieri F, Lutz TA, Giudetti AM, Cassano T, Cifani C, Gaetani S. Oleoylethanolamide decreases frustration stress-induced binge-like eating in female rats: a novel potential treatment for binge eating disorder. Neuropsychopharmacology. 2020 Oct;45(11):1931-1941. doi: 10.1038/s41386-020-0686-z. Epub 2020 Apr 30; Romano A, Gallelli CA, Koczwara JB, Braegger FE, Vitalone A, Falchi M, Micioni Di Bonaventura MV, Cifani C, Cassano T, Lutz TA, Gaetani S. Role of the area postrema in the hypophagic effects of oleoylethanolamide. Pharmacol Res. 2017 Aug;122:20-34; Gaetani S, Fu J, Cassano T, Dipasquale P, Romano A, Righetti L, Cianci S, Laconca L, Giannini E, Scaccianoce S, Mairesse J, Cuomo V, Piomelli D. The fat-induced satiety factor oleoylethanolamide suppresses feeding through central release of oxytocin. J Neurosci. 2010 Jun 16;30(24):8096-101; Fu J, Oveisi F, Gaetani S, Lin E, Piomelli D. Oleoylethanolamide, an endogenous PPAR-alpha agonist, lowers body weight and hyperlipidemia in obese rats. Neuropharmacology. 2005), we decided to perform intraperitoneally (i.p.) administration to rats. The reason for this choice derives from the necessity to have an accurate and controlled dosage of OEA (10mg/kg); in fact, oral administration of substances to laboratory rodents does not always permit to have a controlled and accurate dosage during the administration unless the usage of gavage oral route of administration, which is very stressful for the animals during chronic administrations of substances like our case. In fact, research has shown that dosing of rats and mice via oral gavage can induce significant stress, including increased blood pressure, heart rate, and plasma corticosterone levels (Stress produced by gavage administration in the rat. Brown AP, Dinger N, Levine BS Contemp Top Lab Anim Sci. 2000 Jan; 39(1):17-21; Bonnichsen M, Dragsted M, Hansen AK. The welfare impact of gavaging laboratory rates. Anim Welfare. 2005;14:223–227), thus confounding the interpretation of the results. Moreover, unfortunately, it is not possible to offer OEA to rats dissolved in the water that they drink, since OEA is not soluble in water; nor it is possible to offer OEA mixed in the food they eat because this might cause daily variations of OEA dosage and might interfere with OEA bioavailability. Moreover, adding OEA as a supplement to the food in a chronic paradigm means constantly recalculate in a short period of time the quantity of OEA offered as a function of the average body weights of rats in each group, thus representing an additional bias for our experiments. To the top of our knowledge, one study investigated the effect of oral administration of OEA in mice, given as supplement in the food (Thabuis C, Destaillats F, Landrier JF, Tissot-Favre D, Martin JC. Analysis of gene expression pattern reveals potential targets of dietary oleoylethanolamide in reducing body fat gain in C3H mice. J Nutr Biochem. 2010 Oct;21(10):922-8. doi: 10.1016/j.jnutbio.2009.07.006) and they obtained only a slightly although significant (p<0.05) decrease of food intake over a very long period of treatment corresponding to 4-weeks.

In conclusion, the behavioural and neurochemical effects of OEA in laboratory rodent have been well established by our research group and by others when OEA was administered i.p. and in the present study, we wanted to maintain our previous experimental conditions to operate in a “standard and validated context” to allow a correct interpretation of the results.

However, on the basis of the reviewer’s comments, we added in the text, at the “Materials and methods” section (in the “2.1. Animals, diet and chronic treatments” sub-section) the following sentences: “Theses dosage, vehicle, and route of administration were chosen based on the extensive scientific literature published by ours and other research groups during the last 20 years. The i.p. route of administration was the most reliable to obtain a lower dosage variability, as compared to oral administration of OEA mixed in the rodent diet, and the highest bioavailability of OEA with the less stressful manipulation of the animals.”

Reviewer-Line 107f: Was the efficiency of lipid extraction checked prior to NMR analysis? In LC-MS-based lipidomics approaches for instance (more frequently used than NMR-based), internal standards are usually added to the extraction and measured simultaneously with the analytes in the final sample. How was recovery determined in the present study? This may be decisive in a comparative lipidome study.

Authors: We thank the reviewer for the note. We know that the introduction of an internal standard to measure the efficiency of the extraction procedure is needed when quantitative analysis is made. NMR spectroscopy shows all properties for quantitative analysis: specificity, selectivity, sensitivity, linearity, high dynamic range, response factors unitary, and more importantly, survey universal for all molecules organics that contain protons. The main disadvantage of using the NMR system in quantitative analyses is that the sample must contain at least a reasonably narrow line or a group of signals well separated from all others and you need to know the total number of nuclei represented from the signals used as an internal standard. Thus, the use of an internal standard in the lipid extraction procedure becomes laborious and adds another form of error to the measures. Thus, we bypassed this limit by comparing lipid signals, each of which was reported as a percentage of the total amount of extracted lipids. Thus, the multivariate analysis was made by comparing the relative and not the absolute distribution of total lipid species.

Reviewer-Line 119f: The description of the sphingolipid analysis should be made more precise. Which equal amounts of protein? Which different sphingolipid classes were separated? Which specific standards were used?

Authors: According to the reviewer’s suggestion, in the new version of the manuscript, we indicated the amount of proteins used in the analysis and the classes of standards used for the analysis (paragraph 2.3 Tissue collection and lipid analysis).

Reviewer-Line 148f: Why did the authors only determine the activity of neutral SMase, but not of acid SMase?

Authors: To accomplish the reviewer’s request, in the revised version of the manuscript we also assayed the acid sphingomyelinase activity (paragraph 2.6 “Assay of enzymatic activities” and lines 346-347 in the Result section).

Reviewer-Line 198f: In my opinion, the major limitation of this study is the technique used to determine the hepatic lipidome. Although the NMR analyses performed appear to be accurate, identification of individual lipid classes (that may comprise several hundreds of subspecies) based on chemical shifts of individual structural elements is outdated and not very convincing. Especially because no subspecies could be determined, which e.g. in the case of sphingolipids (fatty acid side chains of different lengths) may exert different biological effects. In general, section 3.1 does not add any novel findings to the field of hepatic lipids

Authors: lines 287-304. According to the referee’s suggestions, it has been better specified that NMR spectroscopy was used to characterized the lipid extracts and identify different lipid classes. Nevertheless, some specific assignments could be performed. As reported in the revised version of the paper, the multiplets at ~5.74 ppm was diagnostic of the characteristic sphingenine moiety vinyl protons (OH-CH-CH=CH-) and has been often assigned to the presence of Sphingolipids including sphingomyelin in selected systems (Barrilero, R.; Gil, M.; Amigó, N.; Dias, C.B.; Wood, L.G.; Garg, M.L.; Ribalta, J.; Heras, M.; Vinaixa, M.; Correig, X. LipSpin: A New Bioinformatics Tool for Quantitative 1H NMR Lipid Profiling. Anal. Chem. 2018, doi:10.1021/acs.analchem.7b04148. Yeboah, F. A., Adosraku, R. K., Nicolaou, A., & Gibbons, W. A. (1995). Proton nuclear magnetic resonance lipid profiling of intact platelet membranes. Annals of clinical biochemistry, 32(4), 392-398). On the other hand, to investigate the increase in sphingomyelin level (suggested by NMR analysis) in the liver of animals treated with OEA, and what was the mechanism behind this increase, we made a TLC analysis of hepatic sphingolipids.

Reviewer-Table 1, line 18: The assigned structural element is also present in e.g. ceramides and sphingosine. How was SM assigned to this signal?

Authors: As reported in the revised version of the paper, Table 1 has been corrected according to the referee’s suggestions.

Reviewer-Line 257: It was not clear to me how the NMR-based semi-quantification was performed. What was the reference (mg tissue, mg protein, another non-lipid signal,..)? Please clarify.

Authors: The NMR-based multivariate analyses allowed the identification of some lipid classes (i.e. PUFA, TAG) as discriminants between the two groups of samples (OEA and VEH). As better clarified and referenced in the revised paper, the relative increase or decrease of the corresponding values for selected buckets, representative of discriminant metabolite signals, is reported in the corresponding normalized box and whisker plots (Chong, J.; Soufan, O.; Li, C.; Caraus, I.; Li, S.; Bourque, G.; Wishart, D.S.; Xia, J. MetaboAnalyst 4.0: Towards more transparent and integrative metabolomics analysis. Nucleic Acids Res. 2018, doi:10.1093/nar/gky310).

Reviewer-Figure 2: The authors should also determine the ceramidase activity to verify that the observed ceramide increase is due to the described OEA-mediated ceramidase inhibition. Without these data, also an OEA-related stimulation of the de novo synthesis could have taken place (which could also have been investigated in the microsomal fractions). What about DAGs? As they are by-products of SM synthesis, their contents would be of interest as well.

Authors: We thank the reviewer for the note. Accordingly, with the reviewer’s request, we assayed ceramidase activity (Figure 2e). Moreover, we also evaluated the level of DAG by a TLC chromatographic method (Fig 2f-h). These new experiments are reported in the Material and method section (paragraphs 2.3 and 2.6) and in the Result section (paragraph 3.3).

Reviewer-Figure 3: An interesting finding in my eyes is the significant reduction of FAS level and activity after OEA treatment. As described by the authors, FAS catalyzes the formation of palmitoyl-CoA. Palmitoyl-CoA is needed to build the sphingoid backbone of sphingolipids (de novo). Furthermore, it is incorporated as a fatty acid side chain in C16:0 Cer and C16:0 SM, abundant subspecies in the liver. The reduction of FAS could therefore rather indicate a reduction of ceramides and subsequent metabolites (such as SM, especially C16:0 subtypes). The opposite appears to be the case. This makes subspecies-resolved lipid analysis even more important, which the authors should aim for in order to improve the manuscript.

Authors: We thank the reviewer for the note because allowed us to ameliorate the knowledge of the relationship between the de novo fatty acid synthesis pathway and sphingolipid de novo synthesis. In the new version of the manuscript, we made an LC-MS/MS analysis of sphingolipids. Surprisingly, although a very strong decrease in de novo fatty acid synthesis rate after OEA administration to rats, we didn’t find significant changes in the amount of C16:0Cer but a significant decrease in the C16:0Glucosylceramide. Obviously, this result deserves a deeper investigation in our future studies.

Reviewer-Lines 334-335: As mentioned above, this should be empirically confirmed.

Authors: According to the reviewer’s request we made additional experiments. Please see our following comment.

Reviewer-Lines 338-350: No experimental data are presented that show direct interconnection between OEA treatment, elevated Cer levels and reduced PPARy expression. This paragraph thus is pure speculation and must be re-written. Preferably, the authors clarify the connection by additional experiments. 

Authors: We agree with the reviewer's consideration. Thus, we have re-written all sentences in which we associated the OEA effect on PPARg with the increase of ceramide. We tried to made experiments to demonstrate such effect of OEA on ceramide level and, in turn, this latter one on PPARg by using hepatic cell lines. However, the results were not so convincing, because, probably due to the cancerous nature of the cultured cells, after OEA addition we found significantly increased level only of sphingomyelin, even in the presence of Carmofur, a well-known inhibitor of ceramidase activity. However, we obtained encouraging data, concerning the effect of both OEA and Carmofur on the PPARg expression that mirrored the in vivo effect. Thus, although we didn’t associate the OEA effect with the increased amount of ceramide, we can obviously associate the effect on PPARg expression on the changes in sphingolipid metabolism. Thus the new experiments are now reported in a new paragraph in the result section and in the new Figure 5 and accordingly, we rewrote sentences on this relationship.

Minor points:

Reviewer-Include the specie studied in abstract and title.

Authors: Done

Reviewer-Figure 1: Quality/visibility should be improved.

Authors: we tried to improve the quality of the figure and we hope that in the current version the quality of the figure was improved.

Reviewer-Figure 2: Please explain LFV and LFO.

Authors: We apologize for the mistake. The LFV and LFO are a wrong indication of VHE and OEA, respectively and both now have been properly substituted.

Reviewer-Figure 4: CD36 is not shown, but stated in the caption.

Authors: We deleted the CD36 word in the caption of the figure.

Reviewer-Line 321: “we found”

Authors: We corrected the mistake

Reviewer 2 Report

The manuscript by Romano et al. examined the effect of chronic oleoylethanolamide treatment on fatty acid and triacylglycerol synthesis in rats. The subject of the study is interesting given the importance of oleoylethanolamide but the experimental design and the analysis of the data should be better explained.

  1. The experimental design lacks key details that would allow for proper evaluation of the experiments. Specifically:
  2. How many rats were used? How were they allocated to the two groups (i.e., was a block design used? How many animals per group?)
  3. Was the body weight of the animals monitored?
  4. How did the authors obtain the results shown in the immunoblot in Fig. 3A? Did the image result from the analysis of a single gel? If this is the case, then how were the individual protein bands obtained? From the image, it seems like the membrane was cut to pieces and each piece was treated with antibodies separately. However, considering the width of the bands and the small difference in the molecular weight of the proteins shown (e.g., Dgat2:43.7 kDa and Scd1:41.6 kDa), it is very unlikely that these proteins separated well enough on the gel to allow for a clean cut between the bands. Showing the molecular weight marker used would help the reader estimate the size of the protein bands
  5. The size of the proteins shown in the immunoblots in Fig. 3 and Fig. 4 should be included in the text. Also, were the antibodies used validated? Some were raised against human proteins and their specificity against the rat proteins should be confirmed
  6. In Fig. 4, CD36 is included in the figure legend but this result is shown in figure 3.

Author Response

Reviewer 2

The manuscript by Romano et al. examined the effect of chronic oleoylethanolamide treatment on fatty acid and triacylglycerol synthesis in rats. The subject of the study is interesting given the importance of oleoylethanolamide but the experimental design and the analysis of the data should be better explained.

The experimental design lacks key details that would allow for proper evaluation of the experiments.

Specifically:

How many rats were used? How were they allocated to the two groups (i.e., was a block design used? How many animals per group?)

Authors: The indication required are now reported in the new version of the manuscript.

Reviewer-Was the body weight of the animals monitored?

Authors: We thank the reviewer for this question. Yes, the body-weight of the animals was monitored during the entire experiment, and the % body weight changes during the 14-day chronic treatment with OEA the results have now been added to the manuscript. Line 111-113, Fig. 1S

Reviewer-How did the authors obtain the results shown in the immunoblot in Fig. 3A? Did the image result from the analysis of a single gel? If this is the case, then how were the individual protein bands obtained? From the image, it seems like the membrane was cut to pieces and each piece was treated with antibodies separately. However, considering the width of the bands and the small difference in the molecular weight of the proteins shown (e.g., Dgat2:43.7 kDa and Scd1:41.6 kDa), it is very unlikely that these proteins separated well enough on the gel to allow for a clean cut between the bands. Showing the molecular weight marker used would help the reader estimate the size of the protein bands

Authors: We thank the reviewer for this note. Of course, all reported membranes of Figure 3 were derived from different separations.

Reviewer-The size of the proteins shown in the immunoblots in Fig. 3 and Fig. 4 should be included in the text. Also, were the antibodies used validated? Some were raised against human proteins and their specificity against the rat proteins should be confirmed.

Authors: According to the reviewer's request, the molecular size of proteins has been reported in both Figures 3 and 4. Regarding the validation of antibodies, all the antibodies used in the manuscript are recommended for rat protein detection, as reported in the time-sheet of each antibody reported in the Material and Method section. Moreover, we have already used all these antibodies in our recently published work on rats (Giudetti et al. FASEB J. 2020 Jul;34(7):9358-9371).

Reviewer-In Fig. 4, CD36 is included in the figure legend but this result is shown in figure 3.

Authors: We corrected this mistake